# Inhibition of Synaptic Glutamate Exocytosis and Prevention of Glutamate Neurotoxicity by Eupatilin from *Artemisia argyi* in the Rat Cortex

**DOI:** 10.3390/ijms232113406

**Published:** 2022-11-02

**Authors:** Cheng-Wei Lu, Chia-Chan Wu, Kuan-Ming Chiu, Ming-Yi Lee, Tzu-Yu Lin, Su-Jane Wang

**Affiliations:** 1Department of Anesthesiology, Far-Eastern Memorial Hospital, New Taipei City 22060, Taiwan; 2Department of Mechanical Engineering, Yuan Ze University, Taoyuan 32003, Taiwan; 3Division of Cardiovascular Surgery, Cardiovascular Center, Far-Eastern Memorial Hospital, New Taipei City 22060, Taiwan; 4Department of Electrical Engineering, Yuan Ze University, Taoyuan 32003, Taiwan; 5Department of Medical Research, Far-Eastern Memorial Hospital, New Taipei City 22060, Taiwan; 6School of Medicine, Fu Jen Catholic University, New Taipei City 24205, Taiwan; 7Research Center for Chinese Herbal Medicine, College of Human Ecology, Chang Gung University of Science and Technology, Taoyuan 33303, Taiwan

**Keywords:** eupatilin, glutamate release, glutamate excitotoxicity, neuroprotection, synaptosome, cortex

## Abstract

The inhibition of synaptic glutamate release to maintain glutamate homeostasis contributes to the alleviation of neuronal cell injury, and accumulating evidence suggests that natural products can repress glutamate levels and associated excitotoxicity. In this study, we investigated whether eupatilin, a constituent of *Artemisia argyi*, affected glutamate release in rat cortical nerve terminals (synaptosomes). Additionally, we evaluated the effect of eupatilin in an animal model of kainic acid (KA) excitotoxicity, particularly on the levels of glutamate and N-methyl-D-aspartate (NMDA) receptor subunits (GluN2A and GluN2B). We found that eupatilin decreased depolarization-evoked glutamate release from rat cortical synaptosomes and that this effect was accompanied by a reduction in cytosolic Ca^2+^ elevation, inhibition of P/Q-type Ca^2+^ channels, decreased synapsin I Ca^2+^-dependent phosphorylation and no detectable effect on the membrane potential. In a KA-induced glutamate excitotoxicity rat model, the administration of eupatilin before KA administration prevented neuronal cell degeneration, glutamate elevation, glutamate-generating enzyme glutaminase increase, excitatory amino acid transporter (EAAT) decrease, GluN2A protein decrease and GluN2B protein increase in the rat cortex. Taken together, the results suggest that eupatilin depresses glutamate exocytosis from cerebrocortical synaptosomes by decreasing P/Q-type Ca^2+^ channels and synapsin I phosphorylation and alleviates glutamate excitotoxicity caused by KA by preventing glutamatergic alterations in the rat cortex. Thus, this study suggests that eupatilin can be considered a potential therapeutic agent in the treatment of brain impairment associated with glutamate excitotoxicity.

## 1. Introduction

Glutamate is the main excitatory neurotransmitter in the central nervous system (CNS) and mediates excitatory neurotransmission [1]. However, at excess levels, glutamate causes excessive activation of glutamate receptors, including α-amino-3-hydroxy-5-methyl-4-isoxazolepropionic acid (AMPA) and NMDA receptors, which leads to Ca^2+^ overloading, protease activation, free radical elevation, mitochondrial dysfunction and eventually neural death [2,3]. This pathological process is termed excitotoxicity and is thought to be involved in several brain pathologies, including ischemia, stroke, epilepsy, and neurodegeneration [4]. Maintaining glutamate homeostasis through the modulation of synaptic glutamate release has been shown to be an effective approach for neuroprotection in glutamate excitotoxicity-related neurological diseases [5,6,7].

Phytochemicals, especially from medicinal plants or herbs, have been an important source for the discovery and development of new agents to treat neurological diseases [8], and numerous phytochemicals that inhibit glutamate levels and associated excitotoxicity have been reported [9,10]. Eupatilin (5,7 dihydroxy 3′,4′,6 trimethoxyflavone; Figure 1A) is a lipophilic flavonoid isolated from *Artemisia argyi* that belongs to the Asteraceae family and is widely used in food, tea and traditional medicine [11,12]. Eupatilin has a variety of biological activities, including antioxidant, anti-inflammatory, immune regulation, antiallergic, anticancer, and cardioprotective properties [13,14,15,16,17,18,19]. Additionally, several studies have shown the beneficial effects of eupatilin in the CNS, which include the alleviation of ischemia-, intracerebral hemorrhage-, or 1-methyl-4-phenyl-1,2,3,6-tetrahydropyridine (MPTP)-induced brain damage and the inflammatory response [17,19,20,21,22], suggesting a protective effect of this natural compound against different CNS injuries.

It is now clear that the excessive release of glutamate plays a large part in the neurotoxicity associated with neurological diseases, and it has been shown that antagonism of glutamate release is potentially neuroprotective [7,10]. Therefore, compounds that inhibit glutamate release have the added potential of reducing neurological injury through inhibition of excitotoxicity, and this property may be an important part of their protective mechanism [23,24]. However, the role of eupatilin in the modulation of the central glutamatergic system has not been studied to date. Therefore, we investigated whether eupatilin affects glutamate release using rat cerebral cortex nerve terminals (synaptosomes), which is a widely used in vitro model for the study of presynaptic modulation of neurotransmitter release [25]. Additionally, we evaluated whether eupatilin prevents the neuronal degeneration and glutamatergic alterations that occur in kainic acid (KA)-treated rats, a well-established animal model of glutamate excitotoxicity [26,27].

## 2. Results

### 2.1. The Effect of Eupatilin on 4-Aminopyridine (4-AP)-Induced Release of Glutamate in Rat Cerebral Cortex Synaptosomes

To investigate whether eupatilin regulates presynaptic glutamate release, purified synaptosomes isolated from the rat cortex were stimulated by 4-AP, a K^+^ channel blocker 4-AP that opens voltage-gated Ca^2+^ channels (VGCCs) and induces the release of glutamate [28]. Figure 1B shows that 4-AP (1 mM) induced a glutamate release of 7.8 ± 0.1 nmol/mg/5 min in synaptosomes incubated in the presence of 1.2 mM CaCl_2_ (*n* = 8). Eupatilin (1 μM, 3 μM, 5 μM, and 10 μM) incubated 10 min prior to 4-AP addition produced a significant depression in the 4-AP-induced release of glutamate without influencing the basal release of glutamate (eupatilin 1 μM, t_12_ = 7.1, *p* < 0.001; eupatilin 3 μM, t_12_ = 16.4, *p* < 0.001; eupatilin 5 μM, t_14_ = 29.1, *p* < 0.001; eupatilin 10 μM, t_10_ = 16.9, *p* < 0.001). This result indicates that eupatilin caused a concentration-dependent inhibition of 4-AP-induced release of glutamate (IC_50_ = 5.1 μM; Figure 1C). As maximal inhibition of glutamate release was observed with 5 μM eupatilin, the subsequent study in synaptosomes was performed using 5 μM eupatilin.

Under synaptosomes incubated in the extracellular Ca^2+^-free medium, 4-AP-induced glutamate release was significantly reduced (F_2,12_ = 711.1; *p* < 0.001; *n* = 5; Figure 1D). In the extracellular Ca^2+^-free medium, however, eupatilin failed to repress 4-AP-induced release of glutamate (*p* = 0.99). In addition, 4-AP-induced release of glutamate was significantly repressed by bafilomycin A1, which blocks vesicular release (F_2,12_ = 928.2; *p* < 0.001; *n* = 5). In the presence of bafilomycin A1, eupatilin failed to produce significant inhibition of the release of glutamate induced by 4-AP (*p* = 0.97; Figure 1D). These results suggest that eupatilin inhibits 4-AP-induced Ca^2+^-dependent vesicular glutamate release.

### 2.2. The Effect of Eupatilin on the 4-AP-Induced Increase in Cytosolic Ca^2+^ Concentration ([Ca^2+^]_C_) and DiSC_3_(5) Fluorescence in Rat Cerebral Cortex Synaptosomes

We next investigated the mechanism responsible for the eupatilin-mediated depression of glutamate release by assessing [Ca^2+^]_C_ and membrane potential in synaptosomes. As shown in Table 1, [Ca^2+^]_C_ in synaptosomes was increased by 4-AP addition. Preincubation with 5 μM eupatilin before 4-AP addition produced a 15% depression of the 4-AP-induced [Ca^2+^]_C_ increase (t_8_ = 9.5, *p* < 0.001; *n* = 5). Basal [Ca^2+^]_C_ was not affected by the presence of 5 μM eupatilin (t_8_ = −0.22, *p* = 0.83). We also assessed the effect of eupatilin on the fluorescence of DiSC_3_(5), a qualitative monitor of membrane potential [29]. As shown in Table 1, 4-AP addition increased DiSC_3_(5) fluorescence. Preincubation with 5 μM eupatilin did not affect either the basal DiSC_3_(5) fluorescence (t_8_ = 1.6, *p* = 0.2) or the 4-AP-induced DiSC_3_(5) fluorescence increase (t_8_ = 0.26, *p* = 0.8; *n* = 5). These results suggest that the observed reduction of 4-AP-induced [Ca^2+^]_C_ by eupatilin was not due to an influence on the synaptosomal membrane potential and subsequent alteration of Ca^2+^ influx [30].

### 2.3. The Effect of Eupatilin in Combination with ω-Conotoxin GVIA or ω-Agatoxin IVA on 4-AP-Induced Release of Glutamate in Rat Cerebral Cortex Synaptosomes

Ca^2+^ influx through VGCCs causes an increase in intraterminal Ca^2+^ and, subsequently, exocytotic release of glutamate [31,32]. Therefore, we investigated the effect of eupatilin on 4-AP-induced release of glutamate in the presence of VGCC blockers (Figure 2). In the presence of the N-type Ca^2+^ channel blocker ω-conotoxin GVIA (2 μM), which by itself inhibited 4-AP-induced release of glutamate by 41% (F_2,12_ = 1600; *p* < 0.001; *n* = 5), 5 μM eupatilin still repressed 4-AP-induced release of glutamate by an additional 48% (*p* < 0.001). However, in the presence of the P/Q-type Ca^2+^ channel blocker ω-agatoxin IVA (0.5 μM), which by itself inhibited 4-AP-induced release of glutamate by 62% (F_2,12_ = 487.8; *p* < 0.001; *n* = 5), no additional inhibition by eupatilin (5 μM) was observed (*p* = 0.83). These data suggest that the inhibitory effect of eupatilin on 4-AP-induced release of glutamate is mediated by P/Q-type rather than N-type VGCC suppression.

### 2.4. The Effect of Eupatilin on Synapsin I Phosphorylation in Rat Cerebral Cortex Synaptosomes

Synapsin I is an important synaptic vesicle-associated phosphoprotein involved in the regulation of neurotransmitter release [33]. Therefore, we investigated the effect of eupatilin on the phosphorylation of synapsin I in synaptosomes, as shown in Figure 3A. Statistical analysis revealed that there were significant differences in synapsin I phosphorylation among the groups (p-synapsin I (Ser62/67), F_2,12_ = 26.4, *p* < 0.001; p-synapsin I (Ser9), F_2,12_ = 21.6, *p* < 0.001; Figure 3B). The addition of 1 mM 4-AP increased synapsin I phosphorylation at Ser62/67 and Ser9 in the presence of 1.2 mM Ca^2+^ (*p* < 0.001). This 4-AP-induced Ca^2+^-dependent synapsin I phosphorylation increase was reduced by the presence of 5 μM eupatilin compared to the 4-AP group (*p* < 0.001), indicating that the inhibitory effect of eupatilin on 4-AP-induced release of glutamate involves a reduction in synapsin I phosphorylation.

### 2.5. The Effect of Eupatilin on Neuronal Degeneration in the Cortex of Rats Injected with KA

We also investigated whether eupatilin exerted a protective action in KA-treated rats, one of the animal models of glutamate excitotoxicity [26]. Rats were pretreated with an intraperitoneal (i.p.) injection of eupatilin 30 min before i.p. injection of KA. 72 h after KA administration, we performed Fluoro-Jade B (FJB) staining to analyze neuronal degeneration in the cortex [34]. As shown in Figure 4A, numerous FJB^+^ cells were found in the prefrontal and entorhinal cortex of KA-treated groups, which was not the case in the dimethylsulfoxide (DMSO)-injected (control) groups, in which there were no FJB^+^ cells observed. In the cortex of KA-treated groups that received a pretreatment of 10 mg/kg eupatilin, a few FJB^+^ cells were also found. However, no FJB^+^ cells were observed in the cortex of KA-treated groups pretreated with 30 mg/kg eupatilin. Statistical analysis revealed that there were significant differences in the number of FJB^+^ cells among the groups (prefrontal cortex, F_3,8_ = 26.9; *p* < 0.001; entorhinal cortex, F_3,8_ = 15.3; *p* < 0.01; Figure 4B). Compared with that in the control group, the number of FJB^+^ cells was increased in the cortex of KA-treated rats (*p* < 0.05). However, in the cortex of KA-treated groups that received pretreatment with 10 or 30 mg/kg eupatilin, the number of FJB^+^ cells was lower than that in the KA-treated group (*p* < 0.05), indicating that eupatilin pretreatment significantly attenuated the increase in the number of FJB^+^ cells induced by KA in the cortex. However, we observed that the protective effects conferred by eupatilin in vivo at 10 mg/kg were not as efficient as at 30 mg/kg. Therefore, we used 30 mg/kg eupatilin for the subsequent in vivo experiments.

### 2.6. The Effect of Eupatilin on the Levels of Glutamate, Glutaminase, and Excitatory Amino Acid Transporters (EAATs) in the Cortex of Rats Injected with KA

Administration of KA causes a marked elevation in glutamate concentrations, which are responsible for neuronal injury or death due to excitotoxicity [26]. The suppression of glutamate levels is considered a means of neuroprotection [23,24]. Therefore, we analyzed the effect of eupatilin on glutamate concentration in the cortex of the groups using high-performance liquid chromatography (HPLC). Statistical analysis indicated that there was a significant difference in the glutamate level in the cortex between the groups after 72 h of KA administration (F_2,12_ = 44.01; *p* < 0.001), as shown in Figure 5A. The level of glutamate in the cortex of the KA group was significantly increased compared to the control group (*p* < 0.001). KA-treated groups that received eupatilin pretreatment showed a significant decrease in the level of glutamate compared to the KA-treated group (*p* < 0.001).

An increase in glutamate production by upregulating glutaminase or a decrease in glutamate reuptake by inhibiting EAATs causes excess glutamate accumulation in the synaptic cleft and excitotoxicity [35,36,37]. Therefore, we explored whether the decreased concentration of glutamate in KA-treated rats that received eupatilin pretreatment was mediated by decreasing glutaminase or increasing EAATs. As shown in Figure 5B, the protein levels of glutaminase, EAAT1, EAAT2 and EAAT3 in the cortex 72 h after KA administration were evaluated using Western blot analysis (EAAT1, F_2,12_ = 37.9, *p* < 0.001; EAAT2, F_2,12_ = 0.001, *p* = 0.00; EAAT3, F_2,12_ = 71.8, *p* < 0.001; Glutaminase, F_2,12_ = 25.2, *p* < 0.001). Statistical analysis showed that the protein level of glutaminase was increased, whereas that of EAAT1 and EAAT3 was decreased in the KA group compared to the control group (*p* < 0.001; Figure 5C). In KA-treated groups that received eupatilin pretreatment, the protein level of glutaminase was decreased, and EAAT1 and EAAT3 were increased compared to the KA group (*p* < 0.001; Figure 5C). No significant differences in EAAT2 protein expression among the groups were observed (*p* = 1; Figure 5C).

### 2.7. The Effect of Eupatilin on the Protein Expression of NMDA Receptor Subunits GluN2A and GluN2B in the Cortex of Rats Injected with KA

Significant alterations in the expression of the NMDA receptor subunits GluN2A and GluN2B have been previously shown in excitotoxic brain injury [38]. Therefore, we used Western blotting to assess whether eupatilin affects in the protein expression levels of GluN2A and GluN2B subunits in cortical tissues 72 h after KA administration (Figure 6A). Significant differences were observed among the groups (GluN2A, F_2,12_ = 282.6, *p* < 0.001; GluN2B, F_2,12_ = 407.9, *p* < 0.001; Figure 6B). Compared with that in the control group, the protein expression of GluN2A was decreased (*p* < 0.001), whereas that of GluN2B was increased in the KA group (*p* < 0.001). However, in KA-treated groups that received eupatilin pretreatment, the protein level of GluN2A was increased, and GluN2B was decreased compared to the KA group (*p* < 0.001).

### 2.8. The Effect of Eupatilin on the Protein Level of Death-Associated Protein Kinase 1 (DAPK1) in the Cortex of Rats Injected with KA

DAPK1, a Ca^2+^/calmodulin-dependent protein kinase II, can phosphorylate GluN2B at S1303 and has been implicated in excitotoxic neuronal cell death [39,40,41]. To assess whether eupatilin protects cortical neurons from KA-induced excitotoxic injury by inhibiting DAPK1, the protein levels of DAPK1 (active), phosphorylated DAPK1 (p-DAPK1) at Ser736 (inactive), and phosphorylated GluN2B (p-GluN2B) at Ser1303 in the cortex 72 h after KA administration were analyzed using Western blot (Figure 7A). Significant differences were observed among the groups (DAPK1, F_2,12_ = 225.7, *p* < 0.001; p-DAPK1, F_2,12_ = 152.9, *p* < 0.001; p-GluN2B, F_2,12_ = 514.3, *p* < 0.001; Figure 7B). Compared with the control group, the protein levels of DAPK1 and p-GluN2B were increased (*p* < 0.001), while p-DAPK1 was decreased in the cortex of KA-treated rats (*p* < 0.001). However, in KA-treated groups pretreated with eupatilin, the protein levels of DAPK1 and p-GluN2B were decreased, and p-DAPK1 was increased compared to the KA group (*p* < 0.001).

## 3. Discussion

Excess glutamate causes glutamate receptor overstimulation and Ca^2+^ overloading, which can lead to neuronal cell damage, and agents that prevent these events are potentially neuroprotective [7]. Phytochemicals are promising candidates for treating glutamate-induced excitotoxicity, and novel therapeutic approaches may arise from constituents of plant sources [9,10]. Eupatilin is a flavonoid extracted from the medical plant *Artemisia argyi*. In the present study, our data shown for the first time a correlation between the neuroprotective property of eupatilin and modulation of the glutamate system in the brain.

### 3.1. Eupatilin Decreases the Release of Glutamate Induced by 4-AP from Rat Cerebral Cortex Synaptosomes by Inhibiting P/Q-Type Ca^2+^ Channels and Synapsin I Phosphorylation

The stimulation of nerve terminals with 4-AP induces Ca^2+^-dependent and Ca^2+^-independent release [28,42]. Ca^2+^-dependent glutamate release is the result of vesicular glutamate release, while Ca^2+^-independent release is due to the reversal of the glutamate transporter [30,43]. We found that eupatilin affected the 4-AP-evoked Ca^2+^-dependent rather than Ca^2+^-independent glutamate release in synaptosomes, since eupatilin failed to significantly inhibit 4-AP-evoked glutamate release in the presence of Ca^2+^-free medium or the vesicular glutamate transporter inhibitor bafilomycin A1. Additionally, we observed a clear inhibition of 4-AP-induced [Ca^2+^]_C_ increase by eupatilin, while eupatilin failed to affect synaptosomal membrane potential, indicating that the inhibitory effect of eupatilin on the 4-AP-evoked Ca^2+^-dependent glutamate release is not a consequence of influencing the synaptosomal membrane potential and consequently altering the Ca^2+^ influx [30].

The influx of Ca^2+^ through presynaptic VGCCs, including N-, P- and Q-type Ca^2+^ channels, is an important process to induce glutamate release from nerve terminals [44,45,46]. We have shown here that eupatilin was ineffective in inhibiting the release of glutamate induced by 4-AP in the presence of the P/Q-type Ca^2+^ channel inhibitor ω-agatoxin IVA. ω-Agatoxin IVA alone produced an inhibition of 62% in rat cortical synaptosomes, which is consistent with previous studies [23,47,48]. In the presence of the N-type Ca^2+^ channel inhibitor ω-conotoxin GVIA, which also caused release inhibition when applied alone, eupatilin still significantly repressed 4-AP-induced release of glutamate. Our data indicate that eupatilin reduces the 4-AP-induced [Ca^2+^]_C_ increase in cortical synaptosomes by P/Q-type Ca^2+^ channel suppression. This reduced Ca^2+^ influx subsequently decreases 4-AP-induced glutamate release.

Our results also show an inhibition of synapsin I Ca^2+^-dependent phosphorylation stimulated by 4-AP in synaptosomes by eupatilin. Synapsin I is a vesicle-associated protein that tethers synaptic vesicles to the actin cytoskeleton and mediates clustering of synaptic vesicles at presynaptic terminals [49]. The phosphorylation of synapsin I on serine residues causes the dissociation of synapsin I from synaptic vesicles and increases the availability of synaptic vesicles to promote glutamate release [33,50,51]. The phosphorylation states of synapsin I are regulated by changes in Ca^2+^ influx through VGCCs in the nerve terminal [52]. Likewise, our results show that the eupatilin-mediated decrease in glutamate release was blocked by suppressing P/Q-type Ca^2+^ channels. Therefore, our data indicate that eupatilin represses P/Q-type Ca^2+^ channels and decreases synapsin I phosphorylation, which may consequently inhibit glutamate release from cortical synaptosomes.

### 3.2. Eupatilin Exerts a Protective Effect against KA-Induced Glutamate Neurotoxicity in the Rat Cortex, and This Effect Is Associated with the Prevention of KA-Induced Glutamatergic Alterations

The inhibition of glutamate release results in decreased postsynaptic excitability, providing one reasonable explanation for the neuroprotective effect [23,24]. Therefore, the reduced glutamate release in synaptosomes may be assumed to decrease neuronal excitability, and thereby contribute to the neuroprotective activity of eupatilin. Supporting this hypothesis, we observed that eupatilin has a neuroprotective effect against KA-induced glutamate toxicity in rats. KA is an excitotoxic substance, and its systemic administration in rodents is widely used as a preclinical model to study excitotoxicity [26]. The excitotoxicity of KA is mediated by increasing glutamate release and overstimulating glutamate receptors resulting in the upregulation of glutaminergic activity, which subsequently causes cell death in several brain regions, including the cerebral cortex [53,54,55,56]. Our results showed that administration of KA at a dose of 15 mg/kg caused elevation of the concentration of glutamate and neuronal degeneration in the rat cortex. These results are in line with those of previous studies [23,55,57]. In addition, reversion of these KA-induced alterations to control levels was observed in the rats pretreated with eupatilin. We hypothesize, therefore, that eupatilin has a profound protective action against KA-induced neurotoxicity in the rat cortex and that the anti-excitotoxic property of eupatilin is linked to its ability to prevent the KA-induced increase in glutamate.

The precise mechanisms by which eupatilin prevents KA-induced glutamate increase remain to be resolved. However, it is evident from this study that the inhibition of glutaminase and preservation of EAAT1/EAAT3 could be involved in mediating this effect. Glutaminase is the primary enzyme for the production of glutamate in neurons, and EAAT is responsible for glutamate uptake in the synaptic cleft, which is critical for maintaining normal levels of synaptic glutamate and limiting the excitotoxicity of glutamate in the CNS [35,36,58]. However, increasing glutamate production by upregulating glutaminase or decreasing glutamate clearance by downregulating EAAT contributes to the elevated level of extracellular glutamate that characterizes excitotoxic brain injury [37,59,60,61,62]. In line with previous findings, our results showed that administration of KA resulted in increased glutaminase and decreased EAAT1/EAAT3 protein levels in the cortex. Pretreatment of rats with eupatilin restored KA-induced glutaminase and EAAT1/EAAT3 alterations to control levels. Therefore, eupatilin may contribute to the prevention of the elevation of glutamate concentrations and excitotoxicity induced by KA in the rat cortex by preserving normal levels of glutaminase and EAAT1/EAAT3.

In addition to pathologically high glutamate levels, excitotoxicity can be induced by the overactivation of NMDA receptors, the mechanism of which involves neuronal death induced by an increase in Ca^2+^ [38]. However, the extent of Ca^2+^ increase varies depending on the subunit composition of the NMDA receptor. In the CNS, the NMDA receptor is most prominently composed of combinations of two GluN1 subunits and two GluN2A and/or GluN2B subunits [63]. Of note, a low level of Ca^2+^ influx has been linked to the GluN2A subunit of NMDAR and mediates cell survival, whereas large Ca^2+^ overload is associated with the GluN2B subunit of NMDA receptor and is involved in cell death [64,65,66]. Therefore, increasing GluN2A expression or inhibiting GluN2B expression is a favorable potential mechanism for alleviating excitotoxic brain injury [67,68]. In this study, we observed that administration of KA decreased the protein level of GluN2A and increased the protein level of GluN2B in the rat cortex, which is consistent with previous studies [69,70]. Furthermore, our results showed that the protein expression level of GluN2A was preserved, while GluN2B was significantly repressed in the cortex of KA-treated rats pretreated with eupatilin compared to those without eupatilin intervention. Therefore, we hypothesize that eupatilin may maintain the normal function of NMDA receptors by regulating the expression of the GluN2A and GluN2B subunits, and this phenomenon may be involved in the neuroprotective effect of eupatilin against KA-induced excitotoxicity.

Although the mechanism by which eupatilin regulates the protein levels of GluN2A and GluN2B remains to be further explored, we focused on DAPK1 in this study. DAPK1 is a unique multidomain serine/threonine kinase that plays a crucial role in neuronal function and cell death [71]. Dephosphorylated DAPK1 is active and has been shown to phosphorylate GluN2B at Ser1303 to regulate the function of NMDA receptors [72]. However, excess glutamate causes intracellular Ca^2+^ increases, which may activate DAPK1 and promote its interaction with GluN2B, resulting in excess Ca^2+^ influx through NMDA receptors and neuronal death [39,40,41]. Furthermore, suppressing DAPK1 attenuates the GluN2B-containing NMDAR-mediated Ca^2+^ increase and protects neurons from excitotoxic damage [39,73]. In this study, we found that administration of KA increased the protein levels of DAPK1 and p-GluN2B and decreased the protein level of p-DAPK1 in the cortex. These KA-induced alterations in DAPK1, p-DAPK1 and p-GluN2B were also returned to control levels by eupatilin pretreatment. Therefore, we hypothesize that eupatilin may prevent DAPK1 activation and decrease GluN2B phosphorylation, leading to the preservation of the normal function of NMDA receptors, which is involved in the prevention of neuronal degeneration in rats with KA-induced excitotoxic injury. How eupatilin prevents DAPK1 activation is not yet clear; however, we found that eupatilin reversed the KA-induced glutamate increase and EAAT1/EAAT3 decrease, resulting in maintenance of normal levels of glutamate. Such an effect could be involved in the preventative effect of eupatilin on KA-induced DAPK1 activation.

Taken together, this study demonstrates that eupatilin depresses glutamate release in rat cortical synaptosomes by repressing P/Q-type Ca^2+^ channels and decreasing synapsin I phosphorylation and attenuated KA-induced excitotoxic injury in cortical neurons through prevention of glutamatergic alterations. This study supports the potential application of *Artemisia argyi*, particularly eupatilin, as a natural therapeutic for glutamate excitotoxicity-related neurological diseases.

## 4. Materials and Methods

### 4.1. Reagents and Antibodies

Eupatilin (CFN90190) was acquired from ChemFaces (Wuhan, Hubei, China). Bafilomycin A1 (#1334) was acquired from Tocris Cookson (Bristol, UK). Fura-2-acetoxymethyl ester (Fura-2-AM), DiSC_3_(5), and antibodies against glutaminase (#701965), p-DAPK1 (Ser736; PA5-105872) were acquired from Invitrogen (Carlsbad, CA, USA). ω-conotoxin GVIA (C-300) and ω-agatoxin IVA (STA-500) were acquired from Alomone (Jerusalem, Israel). Antibodies against p-synapsin I (Ser9; #2311), EAAT3 (#14501), GluN2A (#4205), GluN2B (#4207), p-GluN2B (#71335) and β-actin (#3700) were acquired from Cell Signaling (Beverly, MA, USA). Antibodies against EAAT1 (ab416) and EAAT2 (ab41621) were acquired from Abcam (Cambridge, UK). Horseradish peroxidase-conjugated secondary antibodies (GTX213110-01; GTX213111-01) and p-synapsin I (Ser62/67, GTX82591) antibody were acquired from Genetex (Zeeland, MI, USA). FJB (TR-150FJB) was from Biosensis (Thebarton, Adelaide, Australia). 4-AP (A78403), KA (K0205), antibody against DAPK1 (D217) and general reagents were acquired from Sigma-Aldrich (St. Louis, MO, USA).

### 4.2. Animal Preparation

The experiments were carried out in Sprague Dawley rats (male, 150–200 g) that were purchased from BioLASCO (Taipei, Taiwan) and housed in the animal center of Fu Jen Catholic University. The experimental procedures were in accordance with the guidelines for the care and use of laboratory animals issued by the National Institutes of Health (NIH Publications No. 8023, revised 1996), and they were approved by the Institutional Ethics Committee from Fu Jen Catholic University. The total amount of rats used in the study was 40, of which 10 rats were used for in vitro experiments and 30 rats for in vivo experiments.

### 4.3. Preparation of Synaptosomes

Synaptosomes were prepared as previously described [74,75]. Rats were sacrificed by decapitation, and the cerebral cortex was rapidly removed. After a homogenization step in 0.32 M sucrose, the homogenate was centrifuged at 3000× *g* for 10 min to eliminate nuclei and debris. The supernatant was gently layered on top of a discontinuous Percoll gradient and centrifugated for 7 min at 32,500× *g*. The synaptosomal fraction was collected and centrifugated for 10 min at 27,000× *g*. Synaptosomes were then resuspended in HEPES-buffered medium to a final concentration of approximately 0.5 mg/mL, and processed for glutamate release assay, [Ca^2+^]_C_ determination, membrane potential measurement, or Western blot analysis, respectively.

### 4.4. Glutamate Release from Synaptosomes

Glutamate release was measured by the fluorescence increase due to the production of nicotinamide adenine dinucleotide phosphate (NADPH) in the presence of glutamate dehydrogenase and NADP^+^ [43]. Briefly, synaptosomes were incubated in HEPES-buffered medium containing 20 units of glutamate dehydrogenase, 1 mM NADP^+^, and 1.2 mM CaCl_2_ in a stirred and thermostated cuvette maintained at 37 °C in a Perkin-Elmer LS-55B spectrofluorimeter. After 5 min of incubation, 1 mM 4AP was added to stimulate glutamate release. NADPH fluorescence was excited at 340 nm and monitored at 460 nm (emission wavelength) as a measure of glutamate release. Next, 2 s data points were collected. An exogenous glutamate standard (5 nmol) was applied at the end of each experiment and the fluorescence response used to calculate the released glutamate as nmol glutamate per mg synaptosomal protein (nmol/mg). Release values quoted in the text are levels of glutamate cumulatively released after 5 min depolarization and are indicated as nmol/mg/5 min.

### 4.5. [Ca^2+^]_C_ Determination

The Ca^2+^-sensitive fluorophore Fura-2 was used to assess the [Ca^2+^]_C_ using the Perkin-Elmer spectrofluorimeter as described previously [76]. Briefly, synaptosomes were preincubated in HEPES-buffered medium containing 5 μM fura-2, 0.1 mM Ca^2+^, and 16 μM bovine serum albumin for 30 min at 37 °C, before being centrifuged at 5000× *g* for 30 s in a microfuge and then resuspended in HEPES-buffered medium containing 1.2 mM CaCl_2_. Fluorescence data were obtained at 5 s intervals and the [Ca^2+^]_C_ was calculated according to Grynkiewicz et al. (1985) [77].

### 4.6. Membrane Potential Measurement

The cationic carbocyanine dye DiSC_3_(5) was used to assess membrane potential changes as described previously [29]. Synaptosomes were preincubated in HEPES-buffered medium containing 5 μM DiSC_3_(5) and 1.2 mM CaC1_2_ for 10 min in a stirred and thermostatted cuvette at 37 °C. DiSC_3_(5) fluorescence was measured with the Perkin-Elmer spectrofluorimeter, and data were obtained at 2 s intervals.

### 4.7. Glutamate Excitotoxicity Induced by KA in Rats

Glutamate excitotoxicity was induced in 30 rats by i.p. injection of 15 mg/kg KA. Eight rats that received only DMSO (i.p.) were categorized as control, 8 rats that received only KA (15 mg/kg, i.p.) were denominated the KA group, and 14 rats that received eupatilin (10, 30 mg/kg, i.p.) and KA (15 mg/kg, i.p.) were denominated the eupatilin + KA group. The doses of KA and eupatilin were selected based on previous studies and our pilot study [69,78]. Eupatilin was dissolved in 0.1% DMSO. Rats were treated with eupatilin via i.p. injection, 30 min prior to i.p. injection of l5 mg/kg KA. The rats were sacrificed 72 h after KA administration for immunohistochemical examination, glutamate level determination, and Western blot analysis.

### 4.8. Immunohistochemical Examination

Rats were anesthetized with CO_2_ and perfused transcardially by 0.9% saline and then by 4% paraformaldehyde. Fixed brains were removed, post-fixed in the same fixative overnight, and then transferred to a solution of 30% sucrose for 7 days. Coronal sections, 25 μm, were cut on a freezing microtome and maintained free floating in PBS until stained. FJB staining was performed to analysis the degenerating neurons, as described previously [69]. Briefly, the brain sections were immersed in xylene for 15 min. Subsequently, sections were placed in a gradient ethanol of 85 and 75% for dehydration for 5 min each. Sections were then rinsed in distilled water and incubated in a solution containing 0.06% potassium permanganate for 15 min and rinsed two times in distilled water. Sections were immersed in 0.001% FJB solution for 20 min in the dark. Sections were rinsed three times in distilled water, air-dried for 10 min, and cleared in xylene and coverslipped with DPX. The image was observed under a fluorescence microscope (Zeiss Axioskop 40, Göttingen, Lower Saxony, Germany) using 10× and 20× objectives. The FJB^+^ cells per mm^2^ of the cortex were counted at 20× magnification using the ImageJ (Synoptics, Cambridge, UK).

### 4.9. Measurement of Glutamate Levels in the Rat Cortex

Using HPLC, the concentration of glutamate in brain tissue was determined [79]. Briefly, the rats were sacrificed through decapitation and the cortex (50 mg) were homogenized in 50 mM NH_4_Cl and then were centrifuged for 10 min at 4 °C with 15,000× *g*. The supernatant was filtered through a 0.22 µm membrane filter, and injected to a HPLC machine (HTEC-500, Eicom, Kyoto, Japan). The glutamate concentration was determined using peak areas with an external standard method and were expressed as pg/mg protein in brain tissue.

### 4.10. Western Blot Analysis

Immunoblot analysis was performed on total lysates as previously described [75]. Protein samples from total lysates (30 μg) were loaded onto 10% or 15% SDS-PAGE and transferred to nitrocellulose membranes. Membranes were blocked in Tris-buffered saline-Tween (t-TBS: 20 mM Tris, pH 7.4, 150 mM NaCl, and 0.05% Tween 20) containing 5% non-fat dry milk and 0.1% BSA for 1 h at 4 °C, and then incubated overnight at 4°C with the following primary antibodies: anti-pSer62/67 synapsin I (1:2000), anti-pSer9 synapsin I (1:500), anti-EAAT1 (1:10,000), anti-EAAT2 (1:50,000), anti-EAAT3 (1:10,000), anti-glutaminase (1:10,000), anti-GluN2A (1:3000), anti-GluN2B (1:3000), anti-pSer1303 GluN2B (1:3000), anti-DAPK1 (1:500), anti-pSer736 DAPK1 (1:1000), and anti-β-actin (1:5000). Afterward, the membranes were washed and incubated for 1 h at room temperature with appropriate horseradish peroxidase-linked secondary antibodies (1:10,000). Immunoblots were visualized using the enhanced chemiluminescence system (Amersham Biosciences Corp, Amersham, Buckinghamshire, UK). The density of protein bands was quantitatively analyzed by ImageJ (Synoptics, Cambridge, UK) and normalized to a loading control β-actin.

### 4.11. Data Analysis

The data analysis was performed using GraphPad Prism version 7 (GraphPad Software, Inc. San Diego, CA, USA) and expressed as mean ± SEM. Comparisons between the two groups were analyzed using a two-tailed Student’s t test. Comparisons among multiple groups were analyzed by one-way analysis of variance (ANOVA) and post hoc Tukey’s tests. Differences between groups were considered statistically significant at *p* < 0.05.

## Figures and Tables

**Figure 1 ijms-23-13406-f001:**
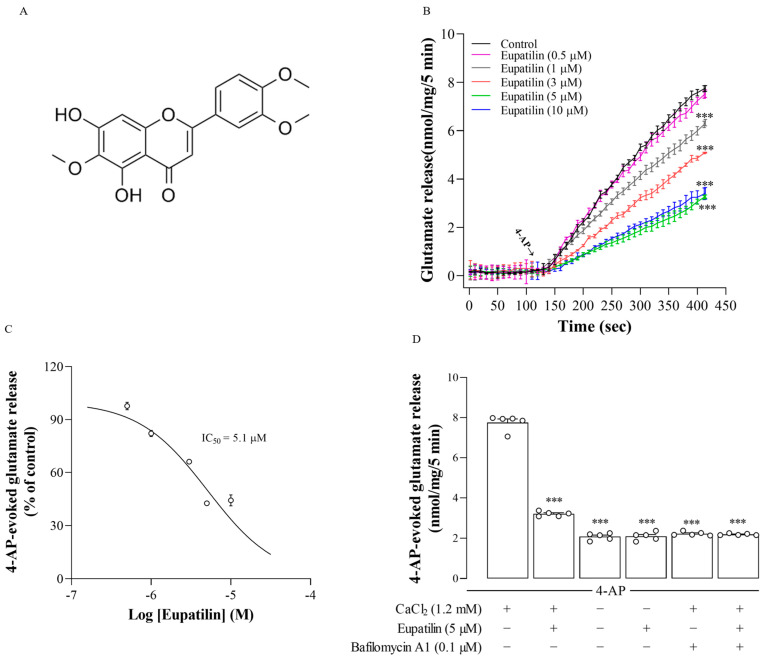
(**A**) The structure of eupatilin. (**B**) 4-AP-induced release of glutamate from rat cerebral cortex synaptosomes in the control and in the presence of eupatilin at concentrations of 0.5, 1, 3, 5, and 10 μM. (**C**) Dose–response curve for eupatilin inhibition of 4-AP-induced release of glutamate. (**D**) The influence of Ca^2+^ free medium or bafilomycin A1 on eupatilin-mediated inhibition of 4-AP-induced glutamate release. Eupatilin or bafilomycin A1 was added 10 min before 4-AP addition. Data are the mean ± SEM. ***, *p* < 0.001, as compared to the control; *n* = 4–8 per group.

**Figure 2 ijms-23-13406-f002:**
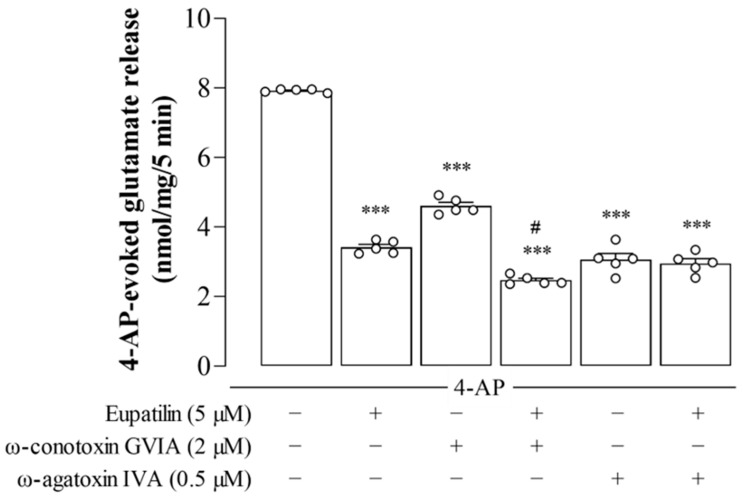
The effect of eupatilin in combination with ω-conotoxin GVIA or ω-agatoxin IVA on 4-AP-induced release of glutamate in rat cerebral cortex synaptosomes. Eupatilin was added 10 min before 4-AP addition and ω-conotoxin GVIA or ω-agatoxin IVA was added 20 min before this. Data are the mean ± SEM. ***, *p* < 0.001, as compared to the control; #, *p* < 0.05, as compared to the ω-conotoxin GVIA-treated group; *n* = 5 per group.

**Figure 3 ijms-23-13406-f003:**
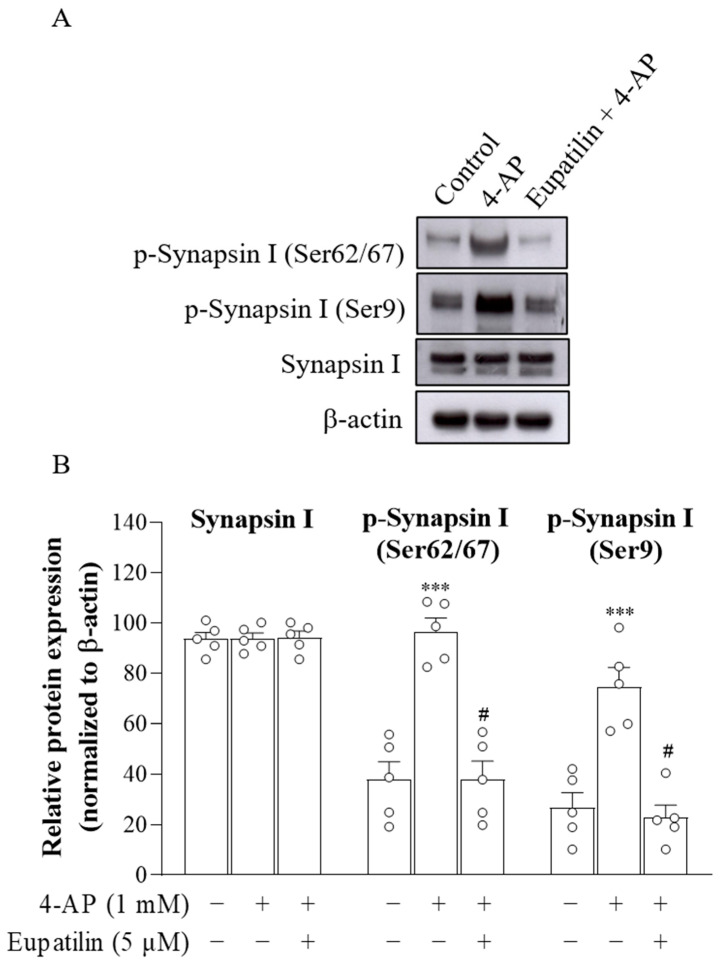
The effect of eupatilin on synapsin I phosphorylation at P-site 1 in rat cerebral cortex synaptosomes. (**A**) Western blot analysis of cortical p-synapsin I expression. (**B**) Densitometric value for p-synapsin I was normalized to β-actin level. Eupatilin was added 10 min before 4-AP addition. Data are the mean ± SEM. ***, *p* < 0.001, as compared to the control; #, p < 0.05, as compared to the 4-AP-treated group; *n* = 5 per group.

**Figure 4 ijms-23-13406-f004:**
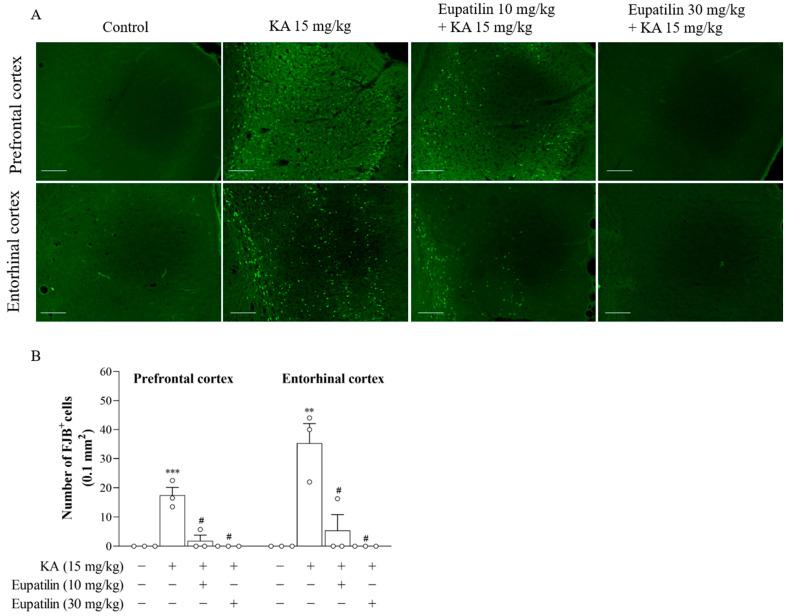
The effect of eupatilin on the neuronal degeneration in the cortex of rats injected with KA. Rats were treated with eupatilin via i.p. injection, 30 min prior to i.p. injection of KA. (**A**) Representative photographs of FJB staining in the prefrontal and entorhinal cortex. Scale bars: 100 μm. Magnification: 200× magnification. (**B**) Quantification of FJB^+^ cells in the prefrontal and entorhinal cortex. Data are the mean ± SEM. ***, *p* < 0.001, **, *p* < 0.01, as compared to the control group; #, *p* < 0.05, as compared to the KA-treated group; *n* = 3 per group.

**Figure 5 ijms-23-13406-f005:**
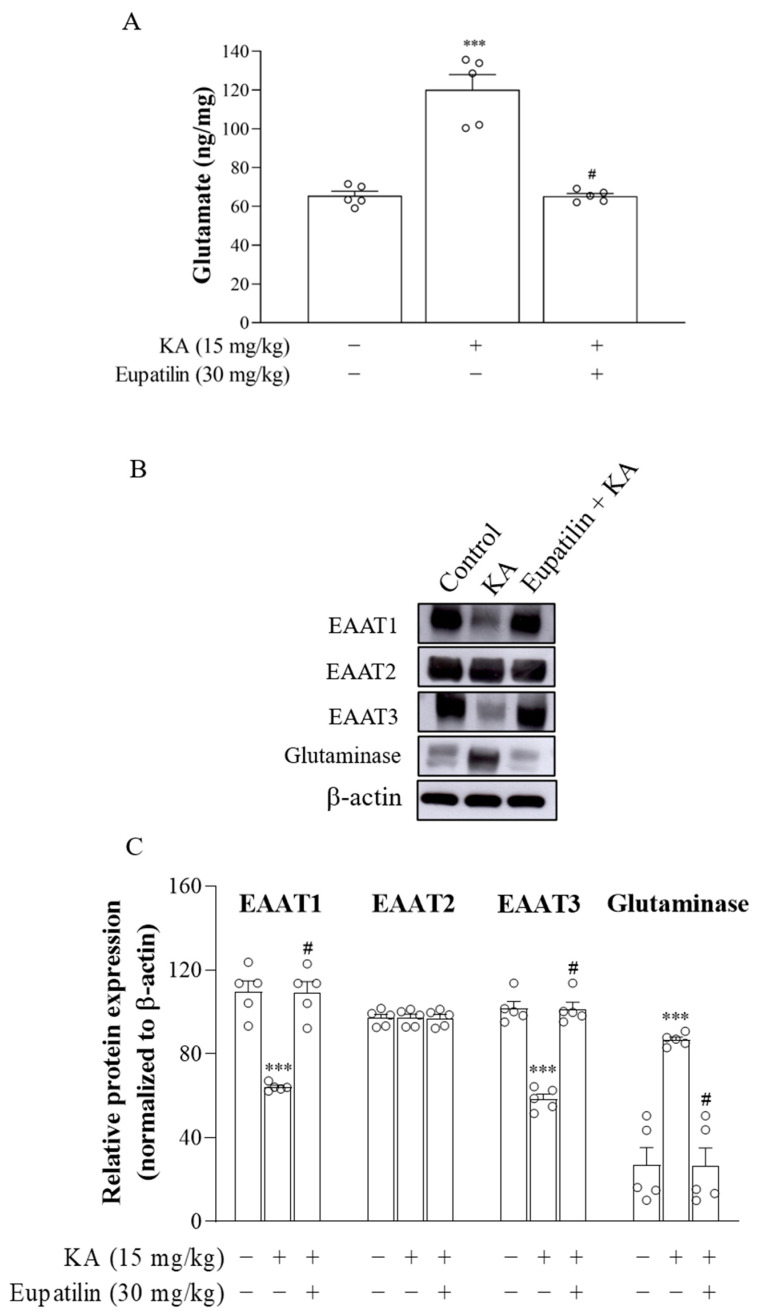
The effect of eupatilin on the levels of glutamate, glutaminase, EAAT1, EAAT2, and EAAT3 in the cortex of rats injected with KA. (**A**) The cortical levels of glutamate were examined by HPLC. (**B**) The protein levels of glutaminase, EAAT1, EAAT2, and EAAT3 in the cortex were analyzed by Western blotting. (**C**) Densitometric values for glutaminase, EAAT1, EAAT2, and EAAT3 were normalized to β-actin levels. Data are the mean ± SEM. ***, *p* < 0.001, as compared to the control group; #, *p* < 0.05, as compared to the KA-treated group; *n* = 5 per group.

**Figure 6 ijms-23-13406-f006:**
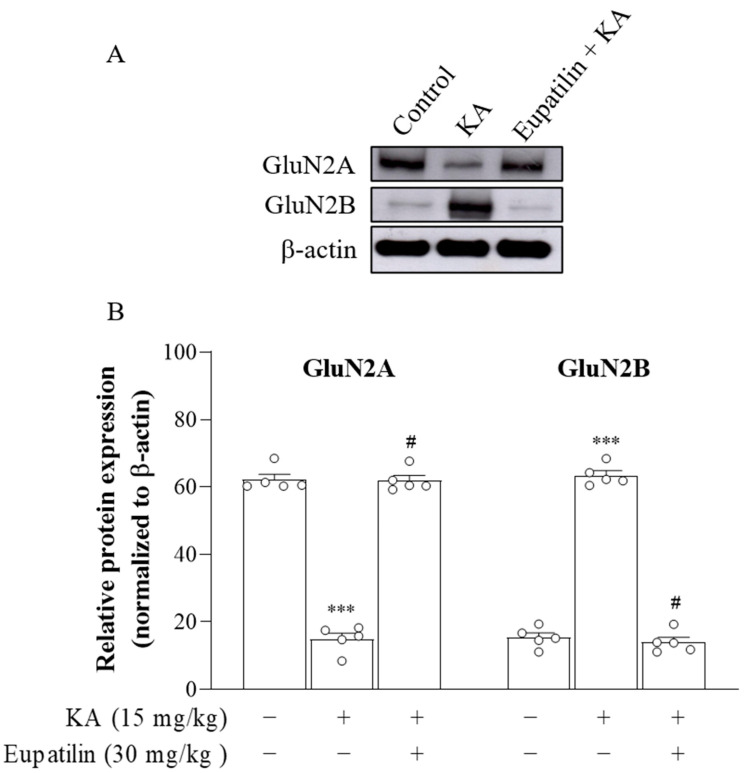
The effect of eupatilin on the levels of GluN2A and GluN2B proteins in the cortex of rats injected with KA. (**A**) Western blot analysis of cortical GluN2A and GluN2B expression. (**B**) Densitometric values for GluN2A and GluN2B were normalized to β-actin level. Data are the mean ± SEM. ***, *p* < 0.001, as compared to the control group; #, *p* < 0.05, as compared to the KA-treated group; *n* = 5 per group.

**Figure 7 ijms-23-13406-f007:**
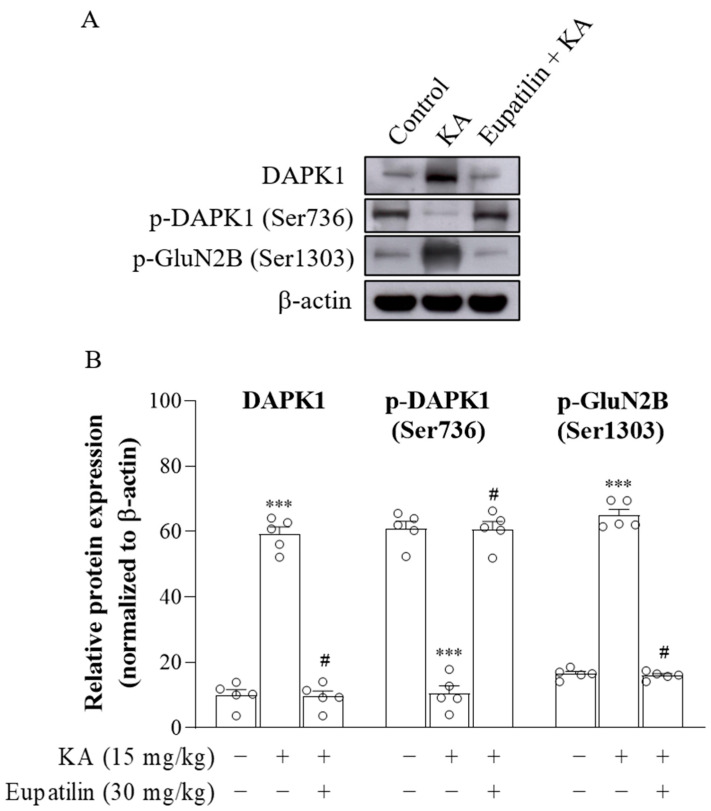
The effect of eupatilin on the protein levels of DAPK1, p-DAPK1 (Ser736) and p-GluN2B (Ser1303) in the cortex of rats injected with KA. (**A**) Western blot analysis of cortical DAPK1 and p-GluN2B expression. (**B**) Densitometric values for DAPK1, p-DAPK1 and p-GluN2B were normalized to β-actin level. Data are the mean ± SEM. ***, *p* < 0.001, as compared to the control group; #, *p* < 0.05, as compared to the KA-treated group; *n* = 5 per group.

**Table 1 ijms-23-13406-t001:** The effect of eupatilin on [Ca^2+^]_C_ and DiSC_3_(5) fluorescence.

	[Ca^2+^]_c_ (nM)		DiSC_3_(5) Fluorescence	
	Basal	4-AP	Basal	4-AP
Control	160.7 ± 1.9	210.5 ± 2.8 ***	0.5 ± 0.1	26.4 ± 0.4 ***
Eupatilin 5 μM	161.2 ± 0.9	179.5 ± 1.6 ^#^	0.4 ± 0.1	26.5 ± 0.4

Data are shown as mean ± SEM. ***, *p* < 0.001, as compare with basal; ^#^, *p* < 0.001, as compare with 4-AP; *n* = 5 per group.

## Data Availability

The data presented in this study are available on request from the corresponding author.

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
