# Peer review of "Inhibition of Synaptic Glutamate Exocytosis and Prevention of Glutamate Neurotoxicity by Eupatilin from Artemisia argyi in the Rat Cortex"

_ijms, 2022, doi:10.3390/ijms232113406_

Round 1

Reviewer 1 Report

The manuscript "Depression of synaptic glutamate exocytosis and prevention of glutamate neurotoxicity by eupatilin from Artemisia argyi in the rat cortex" is high quality paper. Congratulations to the research group.

The manuscript was prepared carefully. The paper can be accepted with minor revisions.

1.      The captions of figure 3 (western blot figure) KA and eupatilin + KA were not matched with the description of adding 1mM of 4-AP. Also in the experimental section, the western blot analysis of synapsin I should be in the 4.3 part. “Experiments with rat cortical synaptosomes”

Author Response

Response to reviewer 1

ijms-1999153R1

We thank the reviewer for the critical comments and constructive suggestions.

  1. The captions of figure 3 (western blot figure) KA and eupatilin + KA were not matched with the description of adding 1mM of 4-AP. Also in the experimental section, the western blot analysis of synapsin I should be in the 4.3 part. “Experiments with rat cortical synaptosomes”.

As suggestion by the reviewer, figure 3 is modified. In addition, the method section is modified (line 407, 431, 439, 445, 455, 470, 478, 494).

Reviewer 2 Report

Investigating the biological activity of new candidates for therapeutic use is still an important research topic, and conducting in vitro/in vivo studies often yields important new knowledge. This is another article prepared by the authors in which a similar research scheme was used.

However, the paper is well-written and has a clear rationale.

Still, I have some questions.

1.       Are the authors sure that the word „depression” in the title and the full text is the appropriate word in this context? Is this word confusing? Wouldn’t it be better to use e.g.„a reduction”?

2.       Avoid starting sentences with a number or abbreviation (Line 408)

3.       Could the authors change the description of Figure 3- May be you should identified the graph as A and Westren Blot as B. Am I right that the Western Blot (in Figure 3) was done in animal studies? Could the authors include this information in the Figure description. There is no information about these results  in 2.4 or 2.5 section.

4.       It is difficult to understand the designation „t12, t14 (…)” in lines 87-88. Could the authors explain?

5.       How many in vitro experiments and repeptitions (of one experiment) were performed? Whether “n” denotes repeptition?

Author Response

Response to reviewer 2

ijms-1999153R1

We thank the reviewer for the critical comments and constructive suggestions.

  1. Are the authors sure that the word „depression” in the title and the full text is the appropriate word in this context? Is this word confusing? Wouldn’t it be better to use e.g.„a reduction”?

As suggestion by the reviewer, in the title, the word is changed to ²inhibition².

  1. Avoid starting sentences with a number or abbreviation (Line 408)

As suggestion by the reviewer, the word is modified (Line 408).  

  1. Could the authors change the description of Figure 3- May be you should identified the graph as A and Westren Blot as B. Am I right that the Western Blot (in Figure 3) was done in animal studies? Could the authors include this information in the Figure description. There is no information about these results in 2.4 or 2.5 section.

As suggestion by the reviewer, Figure 3, 5, 6 and 7 are modified. Figure 3: The effect of eupatilin on synapsin I phosphorylation at P-site 1 in rat cerebral cortex synaptosomes (line 157). Figure 5: The effect of eupatilin on the levels of glutamate, glutaminase, EAAT1, EAAT2, and EAAT3 in the cortex of rats injected with KA (lines 216-217). Figure 6: The effect of eupatilin on the levels of GluN2A and GluN2B proteins in the cortex of rats injected with KA (lines235-236). Figure 7: The effect of eupatilin on the protein levels of DAPK1, p-DAPK1 (Ser736) and p-GluN2B (Ser1303) in the cortex of rats injected with KA (lines255-256). In addition, the sentences in the result section are modified (line 147, 150, 205, 211, 213, 214, 227, 229, 247, 249).

  1. It is difficult to understand the designation „t12, t14 (…)” in lines 87-88. Could the authors explain?

T12 is the degree of freedom (df). It refers to the maximum number of logically independent values, which are values that have the freedom to vary, in the data sample. In a 2-sample t-test, N - 2 is used because there are two parameters to estimate.

For example:

T12 = [n = 8 (control group) + n = 6 (eupatilin 1 mM group)] - 2

  1. How many in vitro experiments and repeptitions (of one experiment) were performed? Whether “n” denotes repeptition?

In order to make the sentence is clearer, the sentence is modified to ² n = 4-8 per group² (line 107) ² n = 5 per group² (line 124,143, 160, 221,239, 259).